# Psychological Difficulties in Children and Adolescents during the COVID-19 Lockdown: The Effects of Spending Free Time with Parents or Using Digital Devices

**DOI:** 10.3390/children10081349

**Published:** 2023-08-04

**Authors:** Anna Di Norcia, Chiara Mascaro, Dora Bianchi, Eleonora Cannoni, Giordana Szpunar, Fiorenzo Laghi

**Affiliations:** Department of Social and Developmental Psychology, Sapienza University of Rome, 00185 Rome, Italy; chiara.mascaro@uniroma1.it (C.M.); dora.bianchi@uniroma1.it (D.B.); eleonora.cannoni@uniroma1.it (E.C.); giordana.szpunar@uniroma1.it (G.S.); fiorenzo.laghi@uniroma1.it (F.L.)

**Keywords:** children, adolescents, psychological difficulties, use of digital devices, time with parents, COVID-19

## Abstract

The present study investigated protective and risk factors for psychological difficulties in children and adolescents during the COVID-19 lockdown. Specifically, the role of increased use of digital devices (DDs) for recreational purposes and the role of free time spent with parents were explored. Furthermore, the possible mediating effect of increased use of DDs in the relationship between free time spent with parents and psychological difficulties was tested. Participants were 4412 Italian children and adolescents, divided into two groups: children aged 6–10 years (n = 2248) and adolescents aged 11–18 years (*n* = 2164). Data were collected in Italy during the national lockdown and questionnaires were completed online by parents, who reported on their children’s habits. The daily use of DDs increased significantly during the lockdown compared to the previous period, in both children and adolescents. Additionally, psychological difficulties increased in both children and adolescents during the lockdown, with a more severe impairment for children. Increased use of DDs during the lockdown positively predicted psychological difficulties (children: beta = 0.18, *p* < 0.001; adolescents: beta = 0.13, *p* < 0.001), while free time spent with parents was protective (children: beta = −13, *p* < 0.001; adolescents: beta = −0.12, *p* < 0.001). For children (but not adolescents), increased use of DDs mediated the effects of free time spent with parents on psychological difficulties (children: beta = −0.01, 95% CI [−0.018, −0.002]; adolescents: beta = −0.003, 95% CI [−0.009, 0.003]). The findings provide new insights for education and research.

## 1. Introduction

The recent coronavirus (COVID-19) pandemic originated in Wuhan, China, in 2019, before spreading throughout the world and forcing entire states to quarantine during the first months of 2020. Italy was one of the first European states to be affected by the virus, and from March to June, the entire nation was in lockdown. Almost all spheres of daily life were severely restricted, so that individuals could leave home for only basic necessities, while commercial enterprises selling unnecessary goods and all recreational, athletic, and educational facilities were closed; moreover, individuals who did not cohabitate were forbidden from meeting one another. Under these conditions, the Italian population endured their longest period of home confinement.

During this period, there was an increase in psychological difficulties among children and adolescents. Many studies reported that the pandemic and lockdown had a strong impact on the emotional and social functioning of young people [1]. Furthermore, a systematic review [2] demonstrated that, as a result of the COVID-19 pandemic, children and adolescents experienced significant disruption in their daily lives, and this condition was a triggering factor for mental illness, including anxiety, depression, and/or stress-related symptoms [2,3]. Specifically, the closure of schools and recreational areas led children and adolescents to experience stress, anxiety, and feelings of helplessness [4]. However, the impact of the lockdown on youths varied according to individual vulnerability factors, such as age and education level, the presence of special needs, and predisposing factors for psychological difficulties [4].

Preliminary studies conducted during the pandemic found that children and adolescents experienced psychological difficulties, including increased irritability, inattention, and a greater need for attachment [4]. Children felt insecure, fearful, and isolated, and suffered from sleep disturbances, nightmares, loss of appetite, agitation, inattention, and separation anxiety [1,5]. Moreover, they were particularly vulnerable to behavioral difficulties, including hyperactivity, conduct problems, externalizing symptoms, and general psychological distress [6,7].

Adolescents were found to experience symptoms of anxiety with respect to both the pandemic and their educational future [2,3]; moreover, they experienced difficulties on a social level, exacerbated by forced social distancing. Teenagers particularly suffered from not being able to see their peers, given the fundamental role played by peer relationships in adolescence, and this condition affected their mood and general well-being. In this respect, the lockdown may have represented a dangerous situation for adolescents, given the delicacy of their developmental phase. Studies have shown that to cope, young people generally increased their use of social media [8] during the lockdown, even though excessive usage has been shown to contribute to long-term mental health problems, such as depression, body dissatisfaction, disordered eating, and health-risk behaviors [9,10,11].

Moreover, many studies reported associations between emotional reactions to COVID-19 and mental health outcomes in youths [1]. Orgilés and colleagues [12] identified numerous changes in children and adolescents due to the lockdown situation, specifically pertaining to emotional states and behaviors, including difficulty concentrating, boredom, irritability, restlessness, nervousness, feelings of loneliness, feelings of discomfort, and worry.

Thus, the present study investigated some protective and risk factors for psychological distress in children and adolescents during the COVID-19 lockdown. Specifically, we explored the increased use of digital devices (DDs) for recreational purposes as a possible risk factor; and free time spent engaging in recreational activities with parents as a possible protective factor, in line with studies on positive parenting [4,13]. Moreover, we expected that, during the COVID-19 lockdown, these two factors may have been associated with psychological difficulties in a complex way, since children may have been encouraged to spend their free time using DDs when their parents were less able to entertain them [14], while adolescents may have been more autonomous in managing their free time [15], even during the lockdown.

### 1.1. Increased Use of Digital Devices as a Risk Factor for Psychological Difficulties

During the COVID-19 closure, children and adolescents increased their use of DDs not only for educational purposes, but also for gaming, shopping, watching movies, using social media, and chatting with peers. In some respects, the use of DDs may counteract social isolation by facilitating contact with peers and relatives. However, it may also negatively influence physical activity levels, exacerbate sedentary lifestyles, and disrupt sleep patterns [16]. Thus, on the one hand, the increased use of DDs during the lockdown may have allowed users to reduce symptoms of stress, anxiety, and depression [17]. However, on the other hand, the excessive use of DDs may have increased psychological difficulties and, over the long term, contributed to the onset of addiction problems [17]. Several studies have also shown that excessive use of DDs in children is associated with obesity, impaired physical and cognitive development, and sleep problems [18,19,20].

Regarding the relationship between the use of DDs and psychological difficulties, several studies have shown that frequent use of DDs can produce many problems in children [21], including irritability, a tendency to cry without cause, sleep disturbance, and severe distractibility [21]. This could be dangerous, especially in young children, as it could trigger more serious problems during development. The use of DDs may also encourage multitasking, which is likely to negatively affect children’s ability to maintain attention over a prolonged period; at the same time, multitasking may result in cognitive overload for children, especially those who are very young, making it impossible to sustain [22].

Regarding adolescents, the systematic review conducted by De Miranda and colleagues [23] indicated that, during the lockdown, time spent using DDs increased, not only for school use but also for leisure. This increase had an overall negative impact on users’ mental health. On the one hand, the use of social media may have allowed adolescents to positively face the lockdown and manage moments of boredom, discomfort, and loneliness [1]. Research has shown that, for adolescents who use social networks, DDs can be beneficial, as they provide access to information and social connection, and they can encourage the development of identity and self-expression [8]. However, at the same time, excessive use of DDs can expose adolescents to several dangers. First, fake news can lead them to believe in misinformation (related to, e.g., COVID-19) and thereby increase their psychological difficulties. Indeed, when adolescents spend significant time passively browsing social media to alleviate boredom, they might encounter a flurry of fake news content, which might exacerbate their anxiety, create a sense of frustration and helplessness, and negatively affect their mood [8]. Second, young people may also come across inappropriate content depicting, for example, peers engaging in risky behaviors [11] or even self-harm [24]. Third, DDs emit blue light, which is a wavelength associated with increased alertness and attention. Blue light also suppresses the release of melatonin, which is a sleep-promoting hormone that is naturally produced by the body in the evening hours. Thus, increased use of DDs throughout the day and evening may increase alertness and delay sleepiness, leading to sleep disturbance [25]. Finally, excessive Internet use is an ascertained risk factor for Internet addiction and gaming disorders to which adolescents are specifically vulnerable [26], and recent studies confirmed the increase in problem Internet use during the pandemic [27,28,29]. Therefore, the misuse of DDs during the pandemic can be considered a risk factor for children’s and adolescents’ mental health. Despite these well-known risks of DDs suggested in the literature [21,24,25], most studies conducted during the COVID-19 pandemic considered increased use of DDs a positive factor for adolescents, counteracting their forced social isolation and allowing them to maintain contact with peers [16]. Thus, the present study aimed at improving our understanding of whether increased use of DDs during the lockdown had different effects on psychological well-being for children versus adolescents.

### 1.2. The Protective Role of Parents against Psychological Difficulties

The amount of time parents spend with children is considered a key barometer of optimal parenting [30]. For parents and children, time spent together is particularly important, as it has been shown to improve family balance and children’s well-being [30]. According to several studies, positive parent–child interactions relate to better adaptation in children, as parental warmth is considered a crucial factor for children to feel accepted [31], and it is linked to their lower levels of depressive symptoms and physical and behavioral problems [32]. Positive parent–child interactions also promote children’s internalization of prosocial norms, which can protect them against risky behaviors [32,33,34,35]. As regards adolescents, according to Branje et al. [36], high-quality parent–child interactions protect against psychological difficulties, especially depressive and anxious symptoms, and this relationship seems to be reciprocal, with benefits for both parents and adolescents. Such positive interactions and the presence of a favorable home environment are fundamental for adolescents’ positive and healthy development in cognitive, linguistic, and socio-emotional fields, as well as their behavior management [32,37].

Parent–child interactions and their effect on psychological well-being have been extensively investigated within the literature on positive parenting. The positive parenting model holds that, by spending time with children, parents can promote positive social behavior by serving as effective models for emotion and behavior management [38]. Indeed, positive parenting is based on the modeling of good parenting behavior, which consists of setting limits while at the same time carefully responding to children’s needs, avoiding harsh punishments, and relying on empathic communication [39]. Furthermore, positive parenting helps children to manage and regulate their emotions, by reducing externalizing behaviors, promoting positive social behavior, and favoring the development of effective conflict mediation strategies. Positive parenting involves sensitivity, responsiveness, care, communication, and empowerment, leading to positive developmental outcomes in children and adolescents. It may also represent a protective factor for mental health in children and adolescents during periods of general difficulty, such as the COVID-19 pandemic [38,40].

During the COVID-19 pandemic, families had to cope with an unprecedented situation in which parents suddenly became the only point of reference for children [40]. At this time, parents protected children by attempting to prevent children’s emotional dysregulation, boosting children’s strengths, and increasing children’s self-confidence [41]. Although parents were exposed to high levels of stress themselves, they could still promote positive emotional functioning in children by reassuring them about the impact of the health emergency. Thus, in times of great stress and uncertainty, a safe family and a healthy home environment can serve as relevant protective factors [42].

Much research has reported that, during the pandemic, parents aimed at protecting children’s psychological well-being by working with them to reorganize routines; helping them respect rules and manage time spent on various activities; at the same time, filtering the news broadcasted by social media, limiting children’s exposure to the news, and trying to provide children with clear and adequate information on COVID-19 [40]. Moreover, younger children required more attention, physical contact, and reassurance [1]. With respect to adolescents, in addition to encouraging better emotion management and modeling appropriate coping strategies, parents may have also established rules to guide their responsible use of DDs [4,40]. At the same time, they may have had to reorganize spaces and schedules to accommodate adolescents’ school and extracurricular commitments [43].

Considering this, the fundamental role played by parents in managing the emotional and behavioral difficulties of children and adolescents during the lockdown may be of particular interest. Some studies have shown that, during the lockdown, the presence of important figures and increased time spent with caregivers were protective factors for psychological well-being in minors [4]. According to Milkie and colleagues [30], time spent with parents encourages the development of socially acceptable behaviors and discourages problematic behaviors, in both children and adolescents. Moreover, quality time spent with parents promotes cognitive and linguistic development, due to the many stimulations children and adolescents receive from their parents [13].

On the other hand, the relevance of spending free time with parents seems to differ between the developmental phases of childhood and adolescence. Differently from children, adolescents usually prefer to spend more time alone in their own space, rather than with their parents [44,45]. In fact, the adolescent period is characterized by a series of changes in relationships with adults and peers. According to theories of developmental psychology [46], adolescents seek progressive autonomy from their parents and tend to affiliate more significantly with peers, in order to develop their adult identities. Therefore, adolescents generally spend less time with their parents and instead concentrate their energy on extra-familial activities that mainly involve peers [43].

Overall, the literature suggests that free time spent with parents may have played a significant role in determining the well-being of children and adolescents during the COVID-19 lockdown. Most likely, it could have been more beneficial for children. Conversely, less direct effects may be expected for adolescents, whose psychological well-being during the lockdown may have been more dependent on their (distanced) relationships with peers. Thus, the present study aimed at exploring whether increased time spent with parents engaging in recreational activities during the lockdown impacted psychological well-being and, in particular, whether it represented a protective factor for psychological difficulties. Additionally, the study explored potential differences in this effect between children and adolescents.

### 1.3. The Present Study

The present study aimed at investigating protective and risk factors for psychological difficulties in children and adolescents during the COVID-19 lockdown. Specifically, it explored the roles of increased use of DDs for recreational purposes and increased free time spent with parents during the lockdown, in determining psychological difficulties in children and adolescents, while controlling for individual differences related to age, gender, and pre-existing psychological problems. Moreover, it aimed at verifying the possible indirect role of free time spent using DDs in the relationship between free time spent with parents and psychological difficulties, in both children and adolescents. Accordingly, the following research hypotheses were explored:

**H1:** 
*Increased use of DDs for recreational purposes during the COVID-19 lockdown would predict more psychological difficulties in both children and adolescents, in light of previous evidence. Children are indeed more vulnerable to cognitive overload due to prolonged excessive exposure to digital media [21]. Moreover, misuse of DDs may expose children and adolescents to inappropriate content (e.g., depictions of risky and aggressive behaviors), with detrimental effects on their psychological well-being [11,24]. However, previous research has not explored whether the impact of increased use of DDs is equivalent for children and adolescents. In fact, recent evidence on adolescents suggests that DDs may have been in some way protective against forced isolation during the COVID-19 lockdown [16].*


**H2:** 
*The quantity of free time spent with parents during the COVID-19 lockdown would protect against psychological difficulties in both children and adolescents, as suggested by studies on positive parenting [4,13]. Time spent with parents in a positive atmosphere (e.g., time spent engaging in shared recreational activities) has been shown to promote positive developmental outcomes [30], thereby representing a protective factor for mental health, especially under difficult circumstances (e.g., the COVID-19 pandemic) [1,40]. However, the literature provides contrasting data on children and adolescents with respect to this factor, suggesting that, during adolescence, relationships with parents undergo rapid and significant changes [43,46]. Thus, free time spent with parents may be less relevant for the psychological well-being of adolescents, relative to children.*


**H3:** 
*The use of DDs would mediate the relationship between free time spent with parents and psychological difficulties during the lockdown, with different effects in children versus adolescents. Children would be more likely than adolescents to spend free time using DDs as a consequence of a lack of parental attention [14,21], which could trigger the development of more psychological difficulties. In contrast, because adolescents tend to determine their free time more independently (especially when parenting is based on autonomy and respect) [15], their time spent with parents would not be directly associated with their use of DDs. Therefore, there would be no mediating effects of DDs usage among adolescents, and the mechanisms leading to their psychological difficulties during the lockdown would differ significantly from those of children.*


## 2. Materials and Methods

### 2.1. Participants and Procedure

The study data came from a broader research project on changes in the life habits of children and adolescents during the national lockdown imposed in Italy to limit the spread of COVID-19. Data collection was conducted in April and May 2020, when the Italian population was constrained at home. Parents filled out an anonymous survey reporting on the life habits of their children, and they completed a questionnaire for each child. Parents were recruited online, and the link to the online survey was distributed via social networking sites (i.e., Facebook and WhatsApp). A preliminary informed consent ensured the complete voluntariness and anonymity of parents’ participation, and questionnaires were completed in 10 min, on average. The only exclusion criterion adopted in this study was the age range of children and adolescents (aged 6 to 18 years).

Initially, 5850 parents were reached by the online invitation message, and they accessed the questionnaire. However, 1438 surveys were removed from the final dataset, because some participants abandoned the survey before completion (dropouts, *n* = 140; completeness rate of 97.61%), and others were excluded because they referred to children and adolescents outside the target age range (excluded, *n* = 1298; 22.18%). Thus, the final study sample referred to 4412 Italian children and adolescents aged 6–18 years (M_age_ = 10.77, SD_age_ = 3.36; 49.0% girls), who were divided into two groups: children aged 6–10 years (*n* = 2248; M_age_ = 7.97, SD_age_ = 1.39; 47.4% girls) and early to late adolescents aged 11–18 years (*n* = 2164; M_age_ = 13.67, SD_age_ = 2.10; 50.6% girls).

As regard the socio-demographic background, parents who completed the online survey were aged 22 to 66 years (M_age_ = 44.78, SD_age_ = 5.57; 88.9% women vs. 11.1% men). Most of them were graduates (40.7%) or had a post-graduate education (23.5%), while the 31.6% completed high school, and the 4.1% had a primary or lower secondary school education level. Most of the responding parents (87.5%) were married and/or cohabiting with a partner, while 10.6% were separated or divorced, and 2% were single parents. Regarding their work condition, 48.4% of interviewed parents were currently working at home due to the lockdown, 12.9% kept going to the workplace, 12.7% have partial smart working, and the remaining 26% were not working.

Regarding missing data management, no missing data were found in the final dataset because the survey was set with mandatory answers; therefore, participants who finalized the survey had necessarily completed all questions, whilst participants who abandoned were not registered. The research and study procedure were approved by the ethics committee of Department of Social and Developmental Psychology, Sapienza University of Rome.

### 2.2. Measures

#### 2.2.1. Individual Information

Parents indicated the gender and age of the child or adolescent for whom they were completing the questionnaire. Individual information of parents (i.e., gender, age, education level, marital status, and work condition) was also collected.

#### 2.2.2. Free Time Spent Using Digital Devices

Five items assessed the amount of daily time each child/adolescent spent using the following DDs for recreational purposes (as perceived by parents): (1) television, (2) PC, (3) smartphone, (4) tablet, and (5) PlayStation. Answers were rated on the following Likert-type scale: 0 (never), 1 (less than 1 h), 2 (1 h), 3 (2 h), 4 (3 h), and 5 (4 or more hours). All items were framed to investigate two periods: prior to the lockdown and during the lockdown period. Mean scores for total daily free time spent using DDs were obtained for each time period. The difference between the two periods was also computed to assess the change in daily free time spent using DDs during the lockdown, in comparison to the previous period.

#### 2.2.3. Free Time Spent with Parents during the Lockdown

Two items evaluated the free time each child/adolescent shared with parents during the lockdown: (1) “During the current COVID-19 lockdown, the leisure activities you share with your son/daughter are…”; and (2) “During the current COVID-19 lockdown, the leisure activities the other parent shares with your son/daughter are…”. Answers were rated on a three-point scale, as follows: 0 (less than before), 1 (equal to before), and 2 (more than before). A mean score was computed for the change in leisure activities shared with both parents during the lockdown.

#### 2.2.4. Psychological Difficulties

Psychological difficulties were investigated using eight items created ad hoc to assess specific problem behaviors in children and adolescents, as perceived by parents: (1) difficulty staying still; (2) difficulty concentrating; (3) nervousness and irritability; (4) a tendency to cry for no reason; (5) difficulty falling asleep; (6) restlessness during sleep, with nocturnal awakenings; (7) food refusal; (8) excessive searching for food. All items were framed to investigate two periods: prior to the lockdown (“Before the COVID-19 lockdown, how often did your son/daughter have the following difficulties?”) and during the lockdown period (“Currently, how often does your son/daughter have the following difficulties?”). Answers were rated on a three-point response scale, as follows: 0 (never), 1 (sometimes), and 2 (often).

This instrument showed good reliability in previous research [47]. In the present study, the psychometric properties were confirmed by good and acceptable levels of reliability in the two versions (Cronbach’s alpha of 0.72 for version during lockdown; 0.60 for version prior to lockdown). The internal consistency was also estimated by the average inter-item correlations, (*r* of 0.24 for the version during lockdown and *r* of 0.15 for the version prior to lockdown), which were in the acceptable ranges (0.15–0.50) according to benchmarks by Clark and Watson [48]. In the retrospective version (i.e., estimating difficulties prior to lockdown), reliability scores were acceptable but lower than in the concurrent assessment (i.e., estimating difficulties during the ongoing lockdown). Therefore, a series of confirmatory factor analyses (CFAs) was also run, in order to confirm the hypothesized two-factor structure of the instrument. We expected a scale composed of two correlated factors (prior vs. during lockdown) with inter-time correlations for the same items. The CFAs were conducted using the LISREL software version 8.80, and the maximum likelihood estimates were computed from the sample correlation matrix. As the Chi-square test statistic is sensitive to large samples size, the goodness of fit of the models was estimated by the root mean square error of approximation and the standardized root mean square residual (RMSEA and SRMR; expected to be < 0.08 in acceptable fit, [49]), and by the comparative fit index, the normed fit index, and the non-normed fit index (CFI, NFI, and NNFI; expected to be >0.90, [50]). Models difference was estimated by the CFI difference (ΔCFI) whose significant values are >0.01. The hypothesized two-factor model showed acceptable fit indexes (RMSEA = 0.068, SRMR = 0.057, CFI = 0.94, NFI = 0.94, and NNFI = 0.92) and explained data significantly better than the more conservative one-factor model (ΔCFI = 0.37), which was unacceptable (RMSEA = 0.16, SRMR = 0.11, CFI = 0.57, NFI = 0.57, and NNFI = 0.45). Thus, the CFAs confirmed the adequacy of the two-factor model to explain our data.

### 2.3. Data Analysis

Two mixed-factor analyses of variance (ANOVAs) were performed with daily free time spent using DDs and psychological difficulties, to investigate changes in these variables between the investigated time periods (prior to vs. during the lockdown), as well as differences between age groups (children vs. adolescents). Age group was entered as a between-subjects factor, and the paired scores of each variable prior to and during the lockdown were entered as within-subjects factor. The results were interpreted through the significance of *p* values–corrected using Bonferroni adjustment in the case of multiple comparisons (critical α of 0.05/4, *p* = 0.012)–and through benchmarks for effect sizes (η^2^_partial_ ≥ 0.01; *d* ≥ 0.20), as suggested by Cohen [51]. A univariate ANOVA was also run to compare children and adolescents on the amount of free time spent with parents during the lockdown. Subsequently, bivariate Pearson’s correlations were computed on the study variables for each age group.

A hierarchical regression analysis was conducted separately for each age group, in order to investigate the predictors of psychological problems during the COVID-19 lockdown in children and adolescents, specifically. In the first step, gender, age, and psychological difficulties prior to the lockdown were entered as control variables. In the second step, the increase in daily free time spent using DDs during the lockdown and the amount of free time spent with parents during the lockdown were added to the regression equation.

A mediation model was then tested using the PROCESS macro for SPSS, version 3.0 [52], Model n. 4, to verify whether the increase in daily use of DDs mediated the relationship between the amount of free time spent with parents and psychological difficulties during the lockdown. The variables were entered as observed indicators and the model was tested separately for each age group. Gender, age, and psychological difficulties prior to the lockdown were also controlled for in each model. Finally, in consideration of the possible bias of cross-sectional data to estimate directionality in mediation models [53], an alternative model was also run to verify the possible significance of the inverse indirect effects, as suggested by scholars [54]. In this inverse model, the psychological difficulties experienced during lockdown were supposed to predict the amount of free time spent with parents, via the mediating effect of the increase in daily use of DDs. Gender, age, and prior psychological difficulties were still controlled as covariates, and the alternative model was tested separately in children and adolescents. Indirect effects and their 95% confidence intervals (CIs) were estimated using bootstrapping with 5000 resamples [50]. CIs were considered significant when they did not include 0 [48].

## 3. Results

### 3.1. Differences Prior to and during the Lockdown in Children and Adolescents

The normal distribution of study variables in the sample was preliminarily ascertained, with skewness and kurtosis values falling in acceptable ranges (±2; [55]). The first mixed-factor ANOVA examined the daily free time spent using DDs in children and adolescents prior to and during the lockdown. The findings showed significant differences for the time period (prior to vs. during the lockdown) and for the age group (children vs. adolescents). Specifically, in both age groups, daily free time spent using DDs significantly increased during the lockdown in comparison to the antecedent period. Furthermore, relative to children, adolescents spent significantly more free time on DDs, both prior to and during the lockdown. No interaction effects were found between time period and age group (see Table 1 for statistics).

The second mixed-factor ANOVA explored psychological difficulties in children and adolescents prior to and during the lockdown. The results were significant for time-period (prior to vs. during lockdown), for age group (children vs. adolescents), and for the interaction between time-period and age group (see Table 1). Multiple comparisons were performed to interpret the direction of this interaction. Three of the four comparisons were significant with small, medium, and large effect sizes, respectively: (1) prior to versus during the lockdown in children, *t*(2247) = −48.01, *p* < 0.001, Cohen *d* = −0.97; (2) prior to versus during the lockdown in adolescents, *t*(2163) = −32.39, *p* < 0.001, Cohen *d* = −0.62; and (3) children versus adolescents during the lockdown, *t*(4410) = 15.54, *p* < 0.001, Cohen *d* = 0.47. Conversely, the fourth comparison obtained a negligible effect size: (4) children versus adolescents prior to the lockdown, *t*(4410) = 6.32, *p* < 0.001, Cohen *d* = 0.19. Therefore, during the COVID-19 lockdown, both children and adolescents reported a significant raise in psychological problems. Moreover, children suffered from significantly more psychological difficulties than adolescents during the lockdown, whilst this difference was not present in the previous period.

The univariate ANOVA examining the amount of free time spent with parents during the lockdown yielded a significant difference between children and adolescents but with a negligible effect size. Descriptive statistics of ANOVA analyses are reported in Table 1. Spearman’s correlations among all study variables are reported in Table 2 for children and in Table 3 for adolescents.

### 3.2. Hierarchical Regression Analyses

The assumptions of the hierarchical multiple regression analyses were preliminarily verified on the study variables, with variance inflation factors (VIF) falling within acceptable ranges (i.e., from 1.00 to 1.05 in the present study). The first regression analysis investigated the predictors of psychological problems during the lockdown in children. Step 1 was significant, explaining 30.4% of the variance. Age emerged as a significant negative predictor, and previous psychological distress as a significant positive predictor of psychological difficulties during the lockdown. Step 2 added a significant 5.1% to the explained variance. Age and previous psychological problems remained significant, and the increase in free time spent using DDs during the lockdown emerged as a significant positive predictor, while the quantity of free time spent with parents during the lockdown showed a significant negative effect. Overall, the model explained 35.5% of the variance in children’s psychological difficulties during the lockdown.

The second regression analysis explored psychological difficulties during the lockdown in adolescents. In step 1, a significant 37.8% of the variance was explained. Age showed a significant negative effect, and previous psychological difficulties demonstrated a significant positive effect on psychological distress during the lockdown. Step 2 added a significant 3.2% to the explained variance. The significant effects of age and previous psychological difficulties were confirmed. Moreover, the increase in free time spent using DDs during the COVID-19 lockdown emerged as a significant positive predictor, while the amount of free time spent with parents during the same period was a significant negative predictor. Overall, the final model explained 41% of the variance in adolescents’ psychological difficulties during the lockdown. Table 4 presents the statistics.

### 3.3. Mediation Models

The first mediation model was tested on the group of children (see Figure 1). Free time spent with parents significantly and negatively predicted the increase in free time spent using DDs. In turn, free time spent using DDs significantly and positively predicted psychological difficulties during the lockdown. Controlling for the effects of the mediator and covariates, the direct effect of free time spent with parents on psychological difficulties was significant and negative. Finally, the indirect path from free time spent with parents to psychological difficulties, via the increase in free time spent using DDs, also emerged as significant and negative, standardized beta = −0.01, *SE* = 0.004, 95% CI [−0.018, −0.002]. Overall, the findings showed the presence of a significant indirect effect of free time spent with parents on psychological distress via the increase in free time spent using DDs, in children. Figure 1 presents the model statistics.

The second mediation model was tested on the group of adolescents (see Figure 2). In this model, free time spent with parents was not related to the increase in free time spent using DDs. An increase in free time spent using DDs was instead a positive and significant predictor of psychological problems during the lockdown. Controlling for the effects of the mediator and covariates, the direct effect of free time spent with parents on psychological difficulties was still significant and negative. Conversely, the indirect path via the increase in free time spent using DDs was not significant, standardized beta = −0.003, *SE* = 0.003, 95% CI [−0.009, 0.003]. Overall, the findings showed that the relationship between free time spent with parents and psychological difficulties in adolescents was not mediated by the increase in free time spent using DDs. Figure 2 presents the model statistics.

### 3.4. Alternative Models

To reduce the possible bias of the cross-sectional design [53], an alternative model was also tested to verify whether the study variables would influence each other also in the inverse direction. The inverse indirect effect of psychological difficulties during lockdown (*X*) on free time spent with parents (*Y*) via the increase in time spent using DDs (*M*) was not significant in the group of children, ab = −0.005, *SE* = 0.006, 95% CI [−0.017, 0.006], as well as in the group of adolescents, ab = 0.001, *SE* = 0.005, 95% CI [−0.009, 0.010]. Therefore, the direction hypothesized in our model appeared to be the most adequate to explain the data, suggesting that the proposed order of variables was meaningful.

## 4. Discussion

The present results shed new light on the impact of the national lockdown in Italy—imposed in the early months of 2020 to control the spread of COVID-19—on the psychological health of Italian children and adolescents. The study analyzed risk and protective factors for psychological difficulties in children and adolescents during the lockdown. In particular, we explored the role of increased use of DDs and free time spent with parents, while controlling for the effects of age, gender, and psychological difficulties experienced prior to the lockdown.

First, the results showed that, in both children and adolescents, daily free time spent using DDs increased significantly during the lockdown, compared to the previous period. Specifically, the study explored the use of DDs for leisure activities, thereby excluding the use of DDs for educational purposes. The findings accord with other studies [12,16] conducted during the pandemic, which have linked the increased use of DDs not only to the introduction of distance learning but also to difficulties related to the lockdown situation (e.g., the impossibility of maintaining in-person connections with peers, increased boredom, and difficulty reconciling parents’ work activities with children’s free time) [12,16].

The results also indicate that, during the lockdown, there was a significant increase in psychological difficulties in comparison to the previous period, in both children and adolescents, probably as a consequence of the imposed isolation. Several studies have shown that people of all ages encountered psychological difficulties during the lockdown [5]. The present results are aligned with previous studies [1,5] which suggest that, during the pandemic, children and adolescents experienced insecurity, fear, and isolation, alongside frequent sleep disturbances, nightmares, loss of appetite, agitation, inattention, and separation anxiety. Moreover, the present study also found that relative to adolescents, children suffered from significantly more psychological difficulties during the lockdown, though this age difference was not present in the previous period. These findings seem to support the claims of some scholars [56,57], that children were more vulnerable to psychological difficulties during the pandemic.

The findings also showed that, during the lockdown, children and adolescents spent a similar amount of free time with their parents. Therefore, no relevant differences emerged between children and adolescents, contrary to expectations of greater parental supervision of children and greater independence of adolescents [45].

Finally, the regression analyses showed that, among both children and adolescents, overall psychological difficulties during the lockdown did not vary by gender. Evidence from the pre-pandemic era suggests that, in general, gender differences in psychological health exist. In particular, internalizing symptoms are more common among females, while externalizing problems are more reported by males [58]. However, the present results suggest that both genders suffered equally from psychological difficulties during the lockdown.

Conversely, age was a significant negative predictor of psychological difficulties, as, during the lockdown, younger (vs. older) children, as well as younger (vs. older) adolescents, presented with greater psychological problems. These results support Via et al. [57], who raised alarms that younger children and adolescents may have been more exposed to psychological difficulties during the pandemic. The reason for this difference could relate to children’s greater need for external self-regulation from caregivers [57], in contrast to adolescents, who are generally more physically and emotionally autonomous [44]. A further explanation for the greater psychological difficulties among younger (vs. older) children and adolescents could derive from the sudden disruption of social contacts for younger participants during the lockdown, while older children and adolescents may have been more able to maintain relationships through social media. Finally, the impact of distance learning may have been more detrimental for the psychological well-being of younger children, who, developmentally, are not yet able to learn in an autonomous and self-directed manner [57].

Previous (i.e., pre-lockdown) psychological difficulties also emerged as a significant positive predictor of psychological difficulties during the lockdown, in both children and adolescents. As expected, participants with more psychological difficulties prior to the pandemic showed more psychological difficulties during the lockdown.

Increased free time spent using DDs during the lockdown was a significant positive predictor of psychological difficulties in both age groups. Therefore, the increased use of DDs during the lockdown could be a risk factor for mental health in both children and adolescents. These results support previous findings about the general effects of excessive use of DDs on the well-being of children and adolescents [59]. Moreover, the findings provide preliminary evidence of the effects associated with the increased use of DDs during the COVID-19 lockdown. Pediatric studies [59] have reported that children and teenagers who generally overuse DDs show greater irritability, a greater tendency to cry, more difficulty sleeping, severe inattention, and less productivity.

More importantly, although both age groups reported detrimental effects of DDs use in relation to psychological difficulties, the weight of this relationship was stronger in children, in comparison with adolescents. This may be explained by adolescents’ more frequent use of DDs even before the pandemic period, while children, in contrast, suddenly found themselves using DDs to fill empty moments. Thus, for children, excessive use of DDs may have been a risk factor which abruptly increased during the pandemic, leading to more severe effects on psychological health in accordance with previous studies [21,22].

The findings also showed that greater amounts of time spent with parents during the lockdown corresponded to more significant decreases in psychological difficulties in both children and adolescents. Therefore, the present study provides the first evidence that, during the COVID-19 lockdown, positive time spent with parents was a protective factor against psychological difficulties, for both children and adolescents. This supports previous research showing that time shared between parents and children improves the family balance and well-being and is thus a key barometer of optimal parenting [30]. Positive interactions between parents and children are linked to better adaptation in children and adolescents; additionally, parental warmth and affection are related to lower levels of depressive symptoms and fewer physical and behavioral problems in children [32].

The present results provide interesting insight into adolescents, specifically. On the basis of the literature [43,46], it was expected that adolescents’ psychological well-being would be more dependent on their positive relationships outside the family, rather than time spent with parents. However, in the context of home confinement with no outside social contact, parents appear to have played a pivotal role in protecting the psychological well-being of both children and adolescents. These findings align with previous studies suggesting that, although adolescents generally seek autonomy from parental figures, positive parent–adolescent relationships characterized by relational intimacy and emotional autonomy can support adolescents’ adjustment and psychological well-being [60].

Finally, as regards the hypothesized mediation model, the present findings show that, for children (but not adolescents), the negative association between free time spent with parents and psychological difficulties was partially mediated by free time spent using DDs. Therefore, one of the reasons children benefited from spending free time with their parents is that this reduced the free time they had available to use DDs, thus minimizing the risk of psychological difficulties associated with excessive use of DDs, as reported in various studies [21,22,59].

A possible explanation for this finding is that, during the lockdown, parents may not have always been available to spend time with their children, due to professional or other commitments. Thus, it is conceivable that parents may have attempted to fill their children’s empty moments by allowing them to use DDs [21]. This would have increased children’s time spent using DDs, thereby exposing them to the risks discussed above. The present findings suggest that the more time parents spent with children during the lockdown, the less time children used DDs for recreational activities, and this protected children’s psychological well-being.

With regard to adolescents, the mediation effect of DD use was not significant. Although time spent with parents was a protective factor and time spent using DDs was a risk factor against psychological difficulties, these variables were not associated with one another. Thus, for adolescents, time spent with parents did not influence time spent using DDs during the lockdown. A possible explanation for this is that, as adolescents are more autonomous than children [45], their free time is not necessarily organized by their parents or filled using DDs, but it can also be dedicated to other activities.

## 5. Limitations and Implications

Despite the novelty of our findings, the study is not exempt from some limitations. First, the exclusive use of a parent-report questionnaire failed to consider the perspectives of children and adolescents, with respect to their use of DDs and perceived psychological difficulties. Second, the survey did not include a standardized measure for assessing psychological difficulties, because it was developed specifically as a short and easy instrument, with the purpose to prevent participant overload and dropout. Third, in our mediation model, the direct effects on the criterion variable were small, albeit still significant, and only a small part of the variance was explained by the main predictors (3% to 5%), whilst most is explained by the prior psychological difficulties. Similarly, the effects of the predictor on the mediator—albeit significant in the children group—were negligible in size (Cohen, 1988); nevertheless, a significant indirect pathway was still found in the group of children. Therefore, despite the relevant contribution of our model, it is conceivable that other different variables may have influenced the psychological well-being of children and adolescents during the pandemic (e.g., parental distress and familial difficulties) and they should be taken under control in future studies. Another limitation is that no information was collected about participants in the pre-pandemic period. Thus, it was not possible to conduct a longitudinal comparison between different time points (i.e., before, during, and after the lockdown). Moreover, the use of cross-sectional (rather than longitudinal) data is often addressed as a possible limitation in mediation models [49]. Notwithstanding, the alternative model results have provided further support to our findings. Finally, the usual limitations of observational studies apply, as no causal relationships could be inferred among the study variables.

Future research should try to analyze whether the newly developed patterns for filling free time during the lockdown changed in months to follow, or whether they remained stable or returned to those of the pre-pandemic era. Furthermore, the present study has implications for the educational field. The results suggest that parents should be educated to spend more free time with children, in order to promote healthy psychological development. Specifically, parents should be discouraged from allowing children to fill their free time using DDs, and they should instead aim to spend more time engaged in recreational activities with children, as this appears to have significant positive value for the psychological well-being of both children and adolescents.

## 6. Conclusions

The present study contributes new knowledge about the impact of the COVID-19 lockdown on the mental health of children and adolescents. In particular, the findings detected increased psychological difficulties in both children and adolescents and showed that increased use of DDs during the lockdown was a relevant risk factor for both age groups. On the other hand, a protective role emerged for the relationship with parents, whereby time spent engaging in recreational activities with parents protected against psychological difficulties during the lockdown. Finally, only for the children group, a significant indirect relationship was found, so that increased free time spent with parents significantly reduced the probability of psychological difficulties via reducing the amount of time children spent using DDs. In conclusion, the protective role of time spent with parents was both relevant per se, and because it prevented the excessive use of DDs.

## Figures and Tables

**Figure 1 children-10-01349-f001:**
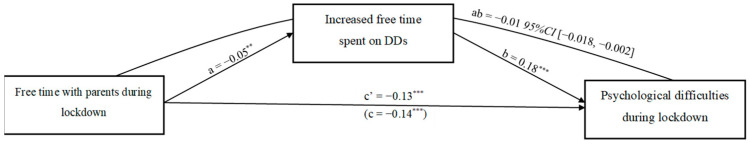
Mediating Effects of the Increase in Free Time Spent Using DDs, Tested in Children. Notes. All relationships in the model are significant. Standardized coefficients are reported. a = effects of free time spent with parents on the mediator; b = effects of the mediator on psychological difficulties; c’ = direct effect of free time spent with parents on psychological difficulties; c = total effect of free time spent with parents on psychological difficulties; ab = indirect effect of free time spent with parents on psychological difficulties via the mediator. *** *p* < 0.001; ** *p* < 0.01.

**Figure 2 children-10-01349-f002:**
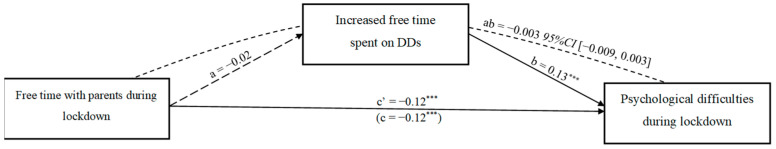
Mediating Effects of the Increase in Free Time Spent Using DDs, Tested in Adolescents. Note. Dotted arrows indicate non-significant relationships. Standardized coefficients are reported. a = effects of free time spent with parents on the mediator; b = effects of the mediator on psychological difficulties; c’ = direct effect of free time spent with parents on psychological difficulties; c = total effect of free time spent with parents on psychological difficulties; ab = indirect effect of free time spent with parents on psychological difficulties via the mediator. *** *p* < 0.001.

**Table 1 children-10-01349-t001:** Differences on study variables divided by time periods and age groups.

	Prior to Lockdown			During Lockdown		
	Children	Adolescents	Children	Adolescents		
	M (SD)	M (SD)	M (SD)	M (SD)	F (df)	*η* ^2^ _partial_
1. Free time spent with parents during the lockdown	--	--	1.47 (0.65)	1.36 (0.64)	Age: 34.39 ***	0.008
					df: 1, 4411	
2. Time spent using DDs	0.78 (0.40)	1.35 (0.56)	1.46 (0.58)	2.07 (0.68)	Time: 8103.09 ***	0.65
					Age: 1541.71 ***	0.26
					Time × *Age*: 3.50	0.001
					df: 1, 4410	
3. Psychological difficulties	0.49 (0.27)	0.45 (0.28)	0.84 (0.41)	0.65 (0.39)	Time: 3267.22 ***	0.43
					Age: 174.75 ***	0.04
					Time × Age: 182.52 ***	0.04
					df: 1, 4410	

Note: *** *p* < 0.001.

**Table 2 children-10-01349-t002:** Descriptive statistics and spearman’s correlations among the study variables for children (*n* = 2248) as perceived by parents.

	1	2	3	4	5	6	7	8	Range	M (SD)
1. Gender (0 = boys; 1 = girls)	1								--	--
2. Age	−0.01	1							6–10	7.97 (1.39)
3. Free time spent with parents during the lockdown	−0.003	−0.04	1						0–2	1.47 (0.65)
4. Time spent using DDs before the lockdown	0.13 ***	0.18 ***	−0.07 **	1					0–5	0.78 (0.40)
5. Time spent using DDs during the lockdown	0.10 ***	0.29 ***	−0.08 ***	0.56 ***	1				0–5	1.46 (0.58)
6. Increase in time spent using DDs	0.01	0.21 ***	−0.05 *	−0.11 **	0.71 ***	1			−5–5	0.68 (0.49)
7. Psychological difficulties before the lockdown	0.09 ***	−0.05 *	−0.04	0.13 ***	0.06 **	−0.03	1		0–2	0.49 (0.27)
8. Psychological difficulties during the lockdown	0.07 **	−0.11 ***	−0.14 ***	0.02	0.12 ***	0.14 ***	0.55 ***	1	0–2	0.84 (0.41)

Note: *** *p* < 0.001; ** *p* < 0.01; * *p* < 0.05.

**Table 3 children-10-01349-t003:** Descriptive statistics and Spearman’s correlations among the study variables for adolescents (*n* = 2164) as perceived by parents.

	1	2	3	4	5	6	7	8	Range	M (SD)
1. Gender (0 = boys; 1 = girls)	1								--	--
2. Age	−0.05 *	1							11–18	13.67 (2.10)
3. Free time spent with parents during the lockdown	−0.02	0.01	1						0–2	1.36 (0.64)
4. Time spent using DDs before the lockdown	0.11 ***	0.27 ***	0.01	1					0–5	1.35 (0.56)
5. Time spent using DDs during the lockdown	0.11 ***	0.07 **	−0.006	0.61 ***	1				0–5	2.07 (0.68)
6. Increase in time spent using DDs	0.03	−0.19 ***	0.01	−0.21 ***	0.59 ***	1			−5–5	0.71 (0.54)
7. Psychological difficulties before the lockdown	0.01	−0.10 ***	−0.07 **	0.07 ***	0.07 **	−0.01	1		0–2	0.45 (0.28)
8. Psychological difficulties during the lockdown	−0.01	−0.17 ***	−0.14***	−0.04	0.07 ***	0.13 ***	0.64 ***	1	0–2	0.65 (0.39)

Note: *** *p* < 0.001; ** *p* < 0.01; * *p* < 0.05.

**Table 4 children-10-01349-t004:** Hierarchical regression analyses for children and adolescents.

	Psychological Difficulties during the Lockdown in Children				Psychological Difficulties during the Lockdown in Adolescents			
	Step 1		Step 2		Step 1		Step 2	
Predictors	beta	SE	beta	SE	beta	SE	beta	SE
Gender (0 = girl; 1 = boy)	0.02	0.01	0.02	0.01	−0.02	0.01	−0.03	0.01
Age	−0.10 ***	0.01	−0.13 ***	0.01	−0.10 ***	0.00	−0.08 ***	0.00
Psychological difficulties before the lockdown	0.54 ***	0.03	0.54 ***	0.03	0.60 ***	0.02	0.59 ***	0.02
Increase in free time spent using DDs	--	--	0.18 ***	0.01	--	--	0.13 ***	0.01
Free time spent with parents during the lockdown	--	--	−0.13 ***	0.01	--	--	−0.12 ***	0.01
Δ*F*	*F* (3, 2244) = 326.05 ***		Δ*F* (2, 2242) = 88.50 ***		*F* (3, 2160) = 437.77 ***		Δ*F* (2, 2158) = 58.12 ***	
*R* ^2^	0.30 ***				0.38 ***			
Δ*R*^2^			0.05 ***				0.03 ***	
Total *R*^2^	0.35 ***				0.41 ***			

Note: *** *p* < 0.001. Standardized beta coefficients are reported.

## Data Availability

The data that support the findings of this study are available from the corresponding author, upon reasonable request.

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
