# Peer review of "Psychological Difficulties in Children and Adolescents during the COVID-19 Lockdown: The Effects of Spending Free Time with Parents or Using Digital Devices"

_children, 2023, doi:10.3390/children10081349_

Round 1

Reviewer 1 Report

This is a simple and clean study that assesses an important issue: the role of parental involvement with children and children’s use of digital devices in children’s well-being during times of COVID.  In a large sample of Italian families (4000+) of preteens and teens, the authors asked parents 15 questions: three related to the types of activities in which children engaged on their digital devices; two on the types of activities in which parents engaged with children; and eight assessing psychological difficulties.  The results were analyzed separately for preteens and teens.  For each group, results showed:

·         Parents who spent more free time with their children tended to have children with fewer psychological difficulties.

·         Children who spent more time using digital devices tended to have higher levels of psychological difficulties.

·         Parents who spent more free time with children tended to have children who spent less time on digital devices (although the relationship between these two variables was weak).

·         The relationship between time spent on devices and psychological difficulties was more pronounced in older than younger children.

Analysis of mediating effects showed that reduced time spent on digital devices mediated the statistically significant indirect relationship between free time parents spent with children and reduced psychological difficulties in younger children, but not with adolescents.

Comments

This study pursues an important thesis, and its findings are theoretically and practically meaningful.  It is well-written, with a clear and extensive literature review.  The methods and analyses are sound.  Beyond that, I have but a few comments.

First, the core finding that that reduced use of digital devices mediates the protective effects of free time spent with parents on children, is a good one, but as far as I can tell, the evidence is weak.  Only a small part of the variance is explained.  More important, if I am interpreting the results correctly, the crucial difference between preteens (a = -.04, significant) and teens (a = -.02, nonsignificant) in the relationship between free time spent with parents and use of digital devices in children, while statistically significant, is very small, and is likely supported by the large sample size.  As a result, the results should be interpreted with this limitation in mind.

Second, it might be helpful to add a third line in Figures 1 and 2 to indicate the indirect relationship between free time spent with parents, use of digital devices and children’s difficulties. It might look something like this:

On page 12, the authors write, “For children, on the other hand, increased use of DDs may not have had this positive aspect, because children are generally not allowed to use social media to contact peers without adult super-vision.”  Is there any evidence to support this claim?

Finally, in this paper, the authors, to their great credit, have been very careful to separate statistical representations of the variables from theoretical/descriptive representations of the variables.  They consistently avoid the pitfall of confusing statistical relationships with contextualized relationships between parents and actual children.  There is one possible exception: in the abstract, the authors write: “For children (but not adolescents), increased use of DDs mediated the negative relationship between free time spent with parents and psychological difficulties.”  The term “negative relationship”, in this context, refers to the negative “statistical” relationship between these variables.  However, this phrase can be read to imply a negative “theoretical” relationship – that is, that free time with parents had a negative effect on (that is, caused an increase in) psychological difficulties.  It would be helpful to clarify this – perhaps by adding the term “statistical” to read “negative statistical relationship” – or clarifying in some other way. 

Author Response

Dear editor,

We are pleased to submit the revised version of the manuscript titled: “Psychological Difficulties in Children and Adolescents During the COVID-19 Lockdown: The Effects of Spending Free Time with Parents or Using Digital Devices” (number children-2492055). We thank the two referees for their suggestions, which helped us to consistently ameliorate the manuscript. We have addressed all suggested changes, and the modified sentences are red colored in the revised manuscript.

Below, detailed responses to the reviewer’s comments are listed point by point.

Reviewer 1:

This study pursues an important thesis, and its findings are theoretically and practically meaningful. It is well-written, with a clear and extensive literature review. The methods and analyses are sound. Beyond that, I have but a few comments.

1.First, the core finding that the reduced use of digital devices mediates the protective effects of free time spent with parents on children, is a good one, but as far as I can tell, the evidence is weak. Only a small part of the variance is explained. More important, if I am interpreting the results correctly, the crucial difference between preteens (a = -.04, significant) and teens (a = - .02, nonsignificant) in the relationship between free time spent with parents and use of digital devices in children, while statistically significant, is very small, and is likely supported by the large sample size. As a result, the results should be interpreted with this limitation in mind.

Answer: We thank the referee for these relevant comments. We have now mentioned these points as limits of our study, in the revised manuscript.

  1. Second, it might be helpful to add a third line in Figures 1 and 2 to indicate the indirect relationship between free time spent with parents, use of digital devices and children’s difficulties. It might look something like this:

Answer: We thank the referee for this interesting suggestion. We have now modified the figures adding a new line indicative of the indirect effects. The statistics in Figure 1 and 2 have also been replaced with standardized regression coefficients.

  1. On page 12, the authors write, “For children, on the other hand, increased use of DDs may not have had this positive aspect, because children are generally not allowed to use social media to contact peers without adult super-vision.” Is there any evidence to support this claim?

Answer: We thank the referee for noting this point. We have decided to remove the whole sentence as this interpretation was not specifically supported by previous studies.

  1. Finally, in this paper, the authors, to their great credit, have been very careful to separate statistical representations of the variables from theoretical/descriptive representations of the variables. They consistently avoid the pitfall of confusing statistical relationships with contextualized relationships between parents and actual children. There is one possible exception: in the abstract, the authors write: “For children (but not adolescents), increased use of DDs mediated the negative relationship between free time spent with parents and psychological difficulties.” The term “negative relationship”, in this context, refers to the negative “statistical” relationship between these variables. However, this phrase can be read to imply a negative “theoretical” relationship – that is, that free time with parents had a negative effect on (that is, caused an increase in) psychological difficulties. It would be helpful to clarify this – perhaps by adding the term “statistical” to read “negative statistical relationship” – or clarifying in some other way.

Answer: We thank the referee for this suggestion. We have now reworded the sentence in the abstract, removing the term “negative relationship”, which might be misleading. The sentence was modified as follows: “For children (but not adolescents), increased use of DDs mediated the effects of free time spent with parents on psychological difficulties”.

Best regards

The Authors

Reviewer 2 Report

It seems that the manuscript entitled “Psychological Difficulties in Children and Adolescents During the COVID-19 Lockdown: The Effects of Spending Free Time with Parents or Using Digital Devices” is a resubmission and the authors have tried their best to improve the quality of the manuscript. The authors have improved the statistical analysis, and the original study design had a good sample size to examine the research question. However, I believe that the present form is not yet ready for publication. Please see my specific comments below.

1. In Abstract, it is unclear how many participants were in each analyzed group. The authors only mentioned the total sample size at 4412 and two groups (parents having children aged 6-10 years and those having adolescents aged 11-18 years). However, it is unclear how many in the children group and how many in the adolescent group.

2. I think that it is essential to provide statistical analysis results (if the authors do not have much space, at least major findings of the statistical analysis results) in Abstract.

3. The Introduction in general is written well. However, when describing the issue of using digital device, I think that the authors should introduce the idea of addictive behavior (or problematic use). There are statements and evidence in the literature to mention the issue of problematic use of internet during COVID-19. The authors may refer to the following references.

https://www.sciencedirect.com/science/article/pii/S0010440X22000372

https://www.sciencedirect.com/science/article/pii/S0010440X21000572

https://doi.org/10.1007/s40429-022-00435-6

https://doi.org/10.1016/j.cobeha.2022.101150

4. The authors did not provide clear information regarding the inclusion and exclusion criteria of the present sample. Therefore, it is hard to interpret “others were excluded because they referred to children and adolescents outside the target age-range (exclusion criterion)”. Also, the response rate should not be 75.4% because the respondents who were not eligible for the recruitment should not be counted as the denominator for response rate.

5. It is unclear to me how the authors could know 5850 parents were invited. If the authors used Facebook and Whatsapp to distribute the online survey, it is hard to know who have entered in the online survey. In this sense, I don’t think that there were 5850 parents were invited. Instead, there were 5850 parents entered in the online survey. In this regard, the term of “response rate” is inaccurate. It should be “completeness rate”. Also, the authors should provide the information regarding how many participants’ data were incomplete, and how many were not used for analysis due to the ineligibility.

6. I wonder why the authors blinded the ethics committee information: “The research and study procedure were approved by the ethics committee of [blinded for peer review].” I don’t think that Children uses masked manuscript review.

7. I believe that the authors’ reference list does not match with the in-text manuscript. For example, the authors said, “This instrument showed good reliability in previous research [43]”. However, reference number 43 is a simulation paper authored by Hu and Bentler. Also, the authors mentioned Clark and Watson [44]. But reference number 44 is not authored by Clark and Watson.

8. The term “confirmative factorial analyses (CFA)” is wrong. It should be “confirmatory factor analysis (CFA)”. If the authors want to use plural, it should be “confirmatory factor analyses (CFAs)”.

9. It is unclear what “retrospective version” means in the sentence, “In the retrospective version, reliability scores were acceptable, but lower than in the concurrent assessment.”

10. In this sentence “Indirect effects and their 95% confidence intervals (CIs) were estimated using bootstrapping with 5,000 samples”, 5000 samples should be corrected to 5000 “resamples”.

11. In the Results section, the authors mentioned the use of Bonferroni method. However, this information should be mentioned in the Data analysis section instead of the Results section.

12. Please provide the information of n for both Tables 2 and 3.

13. I wonder if the authors have the demographic information of the parents (e.g., educational years, gender, age, income). This is important information for the readers to interpret the present study’s findings.

14. In Table 4, it is unclear if the beta is standardized coefficient or unstandardized coefficient.

15. The descriptions “ab = -.01” and “ab = -.003” in the in-text of the Results section are unclear. Specifically, the authors did not introduce “a” and “b” in the main text. Also, it is unclear if this coefficient is standardized or unstandardized.

16. The limitations of the present study should not be under Conclusion section. Please use a separate section of limitations to describe the limitations.

Extensive editing of English language required

Author Response

Dear editor,

We are pleased to submit the revised version of the manuscript titled: “Psychological Difficulties in Children and Adolescents During the COVID-19 Lockdown: The Effects of Spending Free Time with Parents or Using Digital Devices” (number children-2492055). We thank the two referees for their suggestions, which helped us to consistently ameliorate the manuscript. We have addressed all suggested changes, and the modified sentences are red colored in the revised manuscript.

Below, detailed responses to the reviewer’s comments are listed point by point.

Comments of referees

Reviewer 2:

It seems that the manuscript entitled “Psychological Difficulties in Children and Adolescents During the COVID-19 Lockdown: The Effects of Spending Free Time with Parents or Using Digital Devices” is a resubmission and the authors have tried their best to improve the quality of the manuscript. The authors have improved the statistical analysis, and the original study design had a good sample size to examine the research question. However, I believe that the present form is not yet ready for publication. Please see my specific comments below.

  1. In Abstract, it is unclear how many participants were in each analyzed group. The authors only mentioned the total sample size at 4412 and two groups (parents having children aged 6-10 years and those having adolescents aged 11-18 years). However, it is unclear how many in the children group and how many in the adolescent group.

Answer: We thank the referee for this suggestion, and we have modified the abstract adding the number of participants in the two groups.

  1. I think that it is essential to provide statistical analysis results (if the authors do not have much space, at least major findings of the statistical analysis results) in Abstract.

Answer: We thank the referee for this suggestion.We have modified the abstract adding the statistics for the main results.

  1. The Introduction in general is written well. However, when describing the issue of using digital device, I think that the authors should introduce the idea of addictive behavior (or problematic use). There are statements and evidence in the literature to mention the issue of problematic use of internet during COVID-19. The authors may refer to the following references:

https://www.sciencedirect.com/science/article/pii/S0010440X22000372

Savolainen, I., Vuorinen, I., Sirola, A., &Oksanen, A. (2022). Gambling and gaming during COVID-19: The role of mental health and social motives in gambling and gaming problems. Comprehensive Psychiatry, 117, 152331.

https://www.sciencedirect.com/science/article/pii/S0010440X21000572

Gjoneska, B., Potenza, M. N., Jones, J., Corazza, O., Hall, N., Sales, C. M., ... &Demetrovics, Z. (2022). Problematic use of the internet during the COVID-19 pandemic: Good practices and mental health recommendations. Comprehensive psychiatry, 112, 152279.

https://doi.org/10.1007/s40429-022-00435-6

Alimoradi, Z., Lotfi, A., Lin, C. Y., Griffiths, M. D., &Pakpour, A. H. (2022). Estimation of behavioral addiction prevalence during COVID-19 pandemic: a systematic review and meta-analysis. Current addiction reports, 9(4), 486-517.

https://doi.org/10.1016/j.cobeha.2022.101150

Kamolthip, R., Chirawat, P., Ghavifekr, S., Gan, W. Y., Tung, S. E., Nurmala, I., ... & Lin, C. Y. (2022). Problematic Internet use (PIU) in youth: a brief literature review of selected topics. Current Opinion in Behavioral Sciences, 46, 101150.

Answer: We thank the referee for this interesting suggestion, and we have now implemented the literature review in the Introduction section, adding some information aboutthe increased risk for Internet addiction during COVID-19 pandemic. The suggested references have also been added.

  1. The authors did not provide clear information regarding the inclusion and exclusion criteria of the present sample. Therefore, it is hard to interpret “others were excluded because they referred to children and adolescents outside the target age-range (exclusion criterion)”. Also, the response rate should not be 75.4% because the respondents who were not eligible for the recruitment should not be counted as the denominator for response rate.

Answer:We really thank the referee for this and the following comments, that helped us to ameliorate the description of the sampling. We have now added more information about the exclusion criterion of the study (age range), and the “response rate” was modified in “completeness rate”, as suggested below.

  1. It is unclear to me how the authors could know 5850 parents were invited. If the authors used Facebook and Whatsapp to distribute the online survey, it is hard to know who have entered in the online survey. In this sense, I don’t think that there were 5850 parents were invited. Instead, there were 5850 parents entered in the online survey. In this regard, the term of “response rate” is inaccurate. It should be “completeness rate”. Also, the authors should provide the information regarding how many participants’ data were incomplete, and how many were not used for analysis due to the ineligibility.

Answer: We really thank the referee for the comment. We have applied the requested changes in the revised manuscript, as also indicated above. The “response rate” was modified in “completeness rate”, and the percentages of excluded and dropouts were added.

  1. I wonder why the authors blinded the ethics committee information: “The research and study procedure were approved by the ethics committee of [blinded for peer review].” I don’t think that Children uses masked manuscript review.

Answer: Many thanks for this suggestion. The ethic committee is now unblind.

  1. I believe that the authors’ reference list does not match with the in-text manuscript. For example, the authors said, “This instrument showed good reliability in previous research [43]”. However, reference number 43 is a simulation paper authored by Hu and Bentler. Also, the authors mentioned Clark and Watson [44]. But reference number 44 is not authored by Clark and Watson.

Answer: We thank the referee for noting these mistakes. We have now corrected the reference list.

  1. The term “confirmative factorial analyses (CFA)” is wrong. It should be “confirmatory factor analysis (CFA)”. If the authors want to use plural, it should be “confirmatory factor analyses (CFAs)”.

Answer: Many thanks for this suggestion. We have now corrected the sentence as suggested.

  1. It is unclear what “retrospective version” means in the sentence, “In the retrospective version, reliability scores were acceptable, but lower than in the concurrent assessment.”

Answer: We thank the referee for this note. We have now modified the sentence, specifying that the “retrospective version” refers to the psychological difficulties before lockdown (therefore retrospectively reported by parents), and the concurrent assessment refers to the psychological difficulties during the ongoing lockdown.

  1. In this sentence “Indirect effects and their 95% confidence intervals (CIs) were estimated using bootstrapping with 5,000 samples”, 5000 samples should be corrected to 5000 “resamples”.

Answer: Many thanks, we have corrected the sentence as suggested.

  1. In the Results section, the authors mentioned the use of Bonferroni method. However, this information should be mentioned in the Data analysis section instead of the Results section.

Answer: Many thanks, we have modified the text as suggested.

  1. Please provide the information of n for both Tables 2 and 3.

Answer: Many thanks, we have added the number of participants in the title of each table.

  1. I wonder if the authors have the demographic information of the parents (e.g., educational years, gender, age, income). This is important information for the readers to interpret the present study’s findings.

Answer: Many thanks for this relevant suggestion. We have now added the available information about parents, in the description of participants.

  1. In Table 4, it is unclear if the beta is standardized coefficient or unstandardized coefficient.

Answer: Many thanks, we have now specified in the note to Table 4 that the reported regression coefficients are standardized.

  1. The descriptions “ab = -.01” and “ab = -.003” in the in-text of the Results section are unclear. Specifically, the authors did not introduce “a” and “b” in the main text. Also, it is unclear if this coefficient is standardized or unstandardized.

Answer: Many thanks for this relevant note. We have now amended the result section replacing the label “ab” with the term “standardized beta”. Moreover, for clarity, we have specified that standardized regression coefficients are reported, both in regression table, and in the figures (see the Notes to table and figures).

  1. The limitations of the present study should not be under Conclusion section. Please use a separate section of limitations to describe the limitations.

Answer: As suggested, we have now divided the contents in separate sections, as follows: “Limitations and Implications” and “Conclusion”.

  1. Comments on the Quality of English Language: Extensive editing of English language required

Answer: We thank the referee for this note. We have revised the whole manuscript and corrected the English form.

Best regards

The Authors

Round 2

Reviewer 2 Report

The resubmission is much improved. However, some minor issues need to be taken care of.

1. in this description, "albeit significant in the children group—were negligible in size [Cohen, 1988]", the citation format is wrong.

2. "As regardthe socio-demographic background", there is a missing space between "regard" and "the". 

3. "had a post-graduateeducation(23.5%)", there is a missing space between "graduate" and "education".

4. "level.Most", there is a missing space between "." and "Most".

5. "Therefore, a series of confirmatory factor analyses(CFA)" should be "Therefore, a series of confirmatory factor analyses (CFAs)".

6. "DDs.In turn, free time spent using DDssignificantly", this sentence also has missing spaces. 

Should be edited to check for all editorial errors. 

Author Response

We really would like to thank the Reviewer 2 for the careful reading of our manuscript. We have corrected the requested issues, and the all the others editorial errors we were able to find. All the revisions are in red in the text.

Best regards

The Authors